# Fast, Provably convergent IRLS Algorithm for $p$-norm Linear Regression [*]

**Deeksha Adil**
Department of Computer Science
University of Toronto
deeksha@cs.toronto.edu

**Richard Peng**
School of Computer Science
Georgia Institute of Technology
rpeng@cc.gatech.edu

**Sushant Sachdeva**
Department of Computer Science
University of Toronto
sachdeva@cs.toronto.edu

## Abstract

Linear regression in $\ell_p$-norm is a canonical optimization problem that arises in several applications, including sparse recovery, semi-supervised learning, and signal processing. Generic convex optimization algorithms for solving $\ell_p$-regression are slow in practice. Iteratively Reweighted Least Squares (IRLS) is an easy to implement family of algorithms for solving these problems that has been studied for over 50 years. However, these algorithms often diverge for $p > 3$, and since the work of Osborne (1985), it has been an open problem whether there is an IRLS algorithm that is guaranteed to converge rapidly for $p > 3$. We propose $p$-IRLS, the first IRLS algorithm that provably converges geometrically for any $p \in [2, \infty)$. Our algorithm is simple to implement and is guaranteed to find a high accuracy solution in a sub-linear number of iterations. Our experiments demonstrate that it performs even better than our theoretical bounds, beats the standard Matlab/CVX implementation for solving these problems by 10–50x, and is the fastest among available implementations in the high-accuracy regime.

## 1 Introduction

We consider the problem of $\ell_p$-norm linear regression (henceforth referred to as $\ell_p$-regression),

$$\arg \min_{\boldsymbol{x} \in \mathbb{R}^n} \|\boldsymbol{A}\boldsymbol{x} - \boldsymbol{b}\|_p, \tag{1}$$

where $\boldsymbol{A} \in \mathbb{R}^{m \times n}$, $\boldsymbol{b} \in \mathbb{R}^m$ are given and $\|\boldsymbol{v}\|_p = \left(\sum_i |\boldsymbol{v}_i|^p\right)^{1/p}$ denotes the $\ell_p$-norm. This problem generalizes linear regression and appears in several applications including sparse recovery [CT05], low rank matrix approximation [CGK+17], and graph based semi-supervised learning [AL11].

An important application of $\ell_p$-regression with $p \geq 2$ is graph based semi-supervised learning (SSL). Regularization using the standard graph Laplacian (also called a 2-Laplacian) was introduced in the seminal paper of Zhu, Gharamani, and Lafferty [ZGL03], and is a popular approach for graph based SSL, see e.g. [ZBL+04, BMN04, CSZ09, Zhu05]. The 2-Laplacian regularization suffers from degeneracy in the limit of small amounts of labeled data [NSZ09]. Several works have since suggested using the $p$-Laplacian instead [AL11, BZ13, ZB11] with large $p$, and have established its consistency and effectiveness for graph based SSL with small amounts of data [ACR+16, Cal17, RCL19, ST17,

---

[*]Code for this work is available at https://github.com/utoronto-theory/pIRLS.

KRSS15]. Recently, $p$-Laplacians have also been used for data clustering and learning problems [ETT15, EDT17, HFE18]. Minimizing the $p$-Laplacian can be easily seen as an $\ell_p$-regression problem.

Though $\ell_p$-regression is a convex programming problem, it is very challenging to solve in practice. General convex programming methods such as conic programming using interior-point methods (like those implemented in CVX) are very slow in practice. First order methods do not perform well for these problems with $p > 2$ since the gradient vanishes rapidly close to the optimum.

For applications such graph based SSL with $p$-Laplacians, it is important that we are able to compute a solution $\boldsymbol{x}$ that approximates the optimal solution $\boldsymbol{x}^\star$ coordinate-wise rather than just achieving an approximately optimal objective value, since these coordinates determine the labels for the vertices. For such applications, we seek a $(1 + \varepsilon)$-approximate solution, an $\boldsymbol{x}$ such that its objective value, $\|\boldsymbol{Ax} - \boldsymbol{b}\|_p^p$ is at most $(1 + \varepsilon)$ times the optimal value $\|\boldsymbol{Ax}^\star - \boldsymbol{b}\|_p^p$, for some very small $\varepsilon$ ($10^{-8}$ or so) in order achieve a reasonable coordinate-wise approximation. Hence, it is very important for the dependence on $\varepsilon$ be $\log 1/\varepsilon$ rather than $\mathrm{poly}(1/\varepsilon)$. A $\log 1/\varepsilon$ guarantee implies a coordinate-wise convergence guarantee with essentially no loss in the asymptotic running time. Please see the supplementary material for the derivation and experimental evaluation of the coordinate-wise convergence guarantees.

**IRLS Algorithms.** A family of algorithms for solving the $\ell_p$-regression problem are the IRLS (Iterated Reweighted Least Squares) algorithms. IRLS algorithms have been discovered multiple times independently and have been studied extensively for over 50 years e.g. [Law61, Ric64, Osb85, GR97] (see [Bur12] for a detailed survey). The main step in an IRLS algorithm is to solve a weighted least squares ($\ell_2$-regression) problem to compute the next iterate,

$$\boldsymbol{x}^{(t+1)} = \arg\min_{\boldsymbol{x}} (\boldsymbol{Ax} - \boldsymbol{b})^\top \boldsymbol{R}^{(t)} (\boldsymbol{Ax} - \boldsymbol{b}), \tag{2}$$

starting from any initial solution $\boldsymbol{x}^{(0)}$ (usually the least squares solution corresponding to $\boldsymbol{R} = \boldsymbol{I}$). Each iteration can be implemented by solving a linear system $\boldsymbol{x}^{(t+1)} \leftarrow (\boldsymbol{A}^\top \boldsymbol{R}^{(t)} \boldsymbol{A})^{-1} \boldsymbol{A}^\top \boldsymbol{R}^{(t)} \boldsymbol{b}$. Picking $\boldsymbol{R}^{(t)} = \mathrm{diag}\left(|\boldsymbol{Ax}^{(t)} - \boldsymbol{b}|^{p-2}\right)$, gives us an IRLS algorithm where the only fixed point is the minimizer of the regression problem (1) (which is unique for $p \in (1, \infty)$).

The basic version of the above IRLS algorithm converges reliably in practice for $p \in (1.5, 3)$, and diverges often even for moderate $p$ (say $p \geq 3.5$ [RCL19, pg 12]). Osborne [Osb85] proved that the above IRLS algorithm converges in the limit for $p \in [1, 3)$. Karlovitz [Kar70] proved a similar result for an IRLS algorithm with a line search for even $p > 2$. However, both these results only prove convergence in the limit without any quantitative bounds, and assume that you start close enough to the solution. The question of whether a suitable IRLS algorithm converges geometrically to the optimal solution for (1) in a few iterations has been open for over three decades.

**Our Contributions.** We present $p$-IRLS, the first IRLS algorithm that provably converges geometrically to the optimal solution for $\ell_p$-regression for all $p \in [2, \infty)$. Our algorithm is very similar to the standard IRLS algorithm for $\ell_p$ regression, and given an $\varepsilon > 0$, returns a feasible solution $\boldsymbol{x}$ for (1) in $O_p(m^{\frac{p-2}{2(p-1)}} \log \frac{m}{\varepsilon}) \leq O_p(\sqrt{m} \log \frac{m}{\varepsilon})$ iterations (Theorem 3.1). Here $m$ is the number of rows in $\boldsymbol{A}$. We emphasize that the dependence on $\varepsilon$ is $\log \frac{1}{\varepsilon}$ rather than $\mathrm{poly}(\frac{1}{\varepsilon})$.

Our algorithm $p$-IRLS is very simple to implement, and our experiments demonstrate that it is much faster than the available implementations for $p \in (2, \infty)$ in the high accuracy regime. We study its performance on random dense instances of $\ell_p$-regression, and low dimensional nearest neighbour graphs for $p$-Laplacian SSL. Our Matlab implementation on a standard desktop machine runs in at most 2–2.5s (60–80 iterations) on matrices for size $1000 \times 850$, or graphs with 1000 nodes and around 5000 edges, even with $p = 50$ and $\varepsilon = 10^{-8}$. Our algorithm is at least 10–50x faster than the standard Matlab/CVX solver based on Interior point methods [GB14, GB08], while finding a better solution. We also converge much faster than IRLS-homotopy based algorithms [RCL19] that are not even guaranteed to converge to a good solution. For larger $p$, say $p > 20$, this difference is even more dramatic, with $p$-IRLS obtaining solutions with at least 4 orders of magnitude smaller error with the same number of iterations. Our experiments also indicate that $p$-IRLS scales much better than as indicated by our theoretical bounds, with the iteration count almost unchanged with problem size, and growing very slowly (at most linearly) with $p$.

## 1.1 Related Works and Comparison

IRLS algorithms have been used widely for various problems due to their exceptional simplicity and ease of implementation, including compressive sensing [CW08], sparse signal reconstruction [GR97], and Chebyshev approximation in FIR filter design [BB94]. There have been various attempts at analyzing variants of IRLS algorithm for $\ell_p$-norm minimization. We point the reader to the survey by Burrus [Bur12] for numerous pointers and a thorough history.

The works of Osborne [Osb85] and Karlovitz [Kar70] mentioned above only prove convergence in the limit without quantitative bounds and under assumptions on $p$ and that we start close enough. Several works show that it is similar to Newton's method (e.g. [Kah72, BBS94]), or that adaptive step sizes help (e.g. [VB99, VB12]) but do not prove any guarantees.

A few notable works prove convergence guarantees for IRLS algorithms for sparse recovery (even $p < 1$ in some cases) [DDFG08, DDFG10, BL18], and for low-rank matrix recovery [FRW11]. Quantitative convergence bounds for IRLS algorithms for $\ell_1$ are given by Straszak and Vishnoi [SV16b, SV16c, SV16a], inspired by slime-mold dynamics. Ene and Vladu give IRLS algorithms for $\ell_1$ and $\ell_\infty$ [EV19]. However, both these works have poly($1/\varepsilon$) dependence in the number of iterations, with the best result by [EV19] having a total iteration count roughly $m^{1/3}\varepsilon^{-2/3}$.

The most relevant theoretical results for $\ell_p$-norm minimization are Interior point methods [NN94], the homotopy method of Bubeck *et al* [BCLL18], and the iterative-refinement method of Adil *et al.* [AKPS19]. The convergence bounds we prove on the number of iterations required by $p$-IRLS (roughly $m^{\frac{p-2}{2p-2}}$) has a better dependence on $m$ than Interior Point methods (roughly $m^{1/2}$), but marginally worse than the dependence in the work of Bubeck *et al.* [BCLL18] (roughly $m^{\frac{p-2}{2p}}$) and Adil *et al.* [AKPS19] (roughly $m^{\frac{p-2}{2p+(p-2)}}$). Note that we are comparing the dominant polynomial terms, and ignoring the smaller poly($p \log m/\varepsilon$) factors. A follow-up work in this line by a subset of the authors [AS19] focuses on the large $p$ case, achieving a similar running time to [AKPS19], but with linear dependence on $p$. Also related, but not directly comparable are the works of Bullins [Bul18] (restricted to $p = 4$) and the work of Maddison *et al.* [MPT$^+$18] (first order method with a dependence on the condition number, which could be large).

More importantly, in contrast with comparable second order methods [BCLL18, AKPS19], our algorithm is far simpler to implement, and has a *locally greedy* structure that allows for greedily optimizing the objective using a line search, resulting in much better performance in practice than that guaranteed by our theoretical bounds. Unfortunately, there are also no available implementations for any of the above discussed methods (other than interior point methods) in order to make a comparison.

Another line of heuristic algorithms combines IRLS algorithms with a homotopy based approach (e.g. [Kah72]. See [Bur12]). These methods start from a solution for $p = 2$, and slowly increase $p$ multiplicatively, using an IRLS algorithm for each phase and the previous solution as a starting point. These algorithms perform better in practice than usual IRLS algorithms. However, to the best of our knowledge, they are not guaranteed to converge, and no bounds on their performance are known. Rios [Rio19] provides an efficient implementation of such a method based on the work of Rios *et al.* [RCL19], along with detailed experiments. Our experiments show that our algorithm converges much faster than the implementation from Rios (see Section 4).

## 2 Preliminaries

We first define some terms that we will use in the formal analysis of our algorithm. For our analysis we use a more general form of the $\ell_p$-regression problem,

$$\arg \min_{x:Cx=d} \|Ax - b\|_p.\qquad(3)$$

Setting $C$ and $d$ to be empty recovers the standard $\ell_p$-regression problem.

**Definition 2.1** (Residual Problem). *The residual problem of* (3) *at $x$ is defined as,*

$$\max_{\Delta:C\Delta=0} \quad g^\top A\Delta - 2p^2\Delta^\top A^\top RA\Delta - p^p\|A\Delta\|_p^p.$$

*Here $R = diag\left(|Ax - b|^{p-2}\right)$ and $g = pR(Ax - b)$ is the gradient of the objective at $x$. Define $\gamma(\Delta)$ to denote the objective of the residual problem evaluated at $\Delta$.*

**Definition 2.2** (Approximation to the Residual Problem). *Let $\kappa \geq 1$ and $\Delta^\star$ be the optimum of the residual problem. A $\kappa$-approximate solution to the residual problem is $\widetilde{\Delta}$ such that $C\widetilde{\Delta} = 0$, and $\gamma(\widetilde{\Delta}) \geq \frac{1}{\kappa}\gamma(\Delta^\star)$.*

## 3 Algorithm and Analysis

---
**Algorithm 1** $p$-IRLS Algorithm
---
1: **procedure** $p$-IRLS$(\boldsymbol{A}, \boldsymbol{b}, \varepsilon, \boldsymbol{C}, \boldsymbol{d})$
2: $\quad \boldsymbol{x} \leftarrow \arg\min_{\boldsymbol{Cx=d}} \|\boldsymbol{Ax} - \boldsymbol{b}\|_2^2$.
3: $\quad i \leftarrow \|\boldsymbol{Ax} - \boldsymbol{b}\|_p^p / 16p$
4: $\quad$ **while** $\frac{\varepsilon}{16p(1+\varepsilon)} \|\boldsymbol{Ax} - \boldsymbol{b}\|_p^p < i$ **do**
5: $\quad\quad \boldsymbol{R} \leftarrow |\boldsymbol{Ax} - \boldsymbol{b}|^{p-2}$
6: $\quad\quad \boldsymbol{g} = p\boldsymbol{R}(\boldsymbol{Ax} - \boldsymbol{b})$
7: $\quad\quad s \leftarrow \frac{1}{2}i^{(p-2)/p}m^{-(p-2)/p}$
8: $\quad\quad \widetilde{\Delta} \leftarrow \arg\min_{\boldsymbol{g}^\top \boldsymbol{A}\Delta = i/2, \boldsymbol{C}\Delta=0} \Delta^\top \boldsymbol{A}^\top (\boldsymbol{R} + s\boldsymbol{I})\boldsymbol{A}\Delta$
9: $\quad\quad \alpha \leftarrow \text{LINESEARCH}(\boldsymbol{A}, \boldsymbol{b}, \boldsymbol{x}^{(t)}, \widetilde{\Delta})$ $\qquad\qquad \triangleright \alpha = \arg\min_\alpha \|\boldsymbol{A}(\boldsymbol{x} - \alpha\widetilde{\Delta}) - \boldsymbol{b}\|_p^p$
10: $\quad\quad \boldsymbol{x}^{(t+1)} \leftarrow \boldsymbol{x}^{(t)} - \alpha\widetilde{\Delta}$
11: $\quad\quad$ **if** $\text{INSUFFICIENTPROGRESSCHECK}(\boldsymbol{A}, \boldsymbol{R} + s\boldsymbol{I}, \widetilde{\Delta}, i)$ **then** $i \leftarrow i/2$
12: $\quad$ **return** $\boldsymbol{x}$
---

---
**Algorithm 2** Check Progress
---
1: **procedure** $\text{INSUFFICIENTPROGRESSCHECK}(\boldsymbol{A}, \boldsymbol{R}, \Delta, i)$
2: $\quad \lambda \leftarrow 16p$
3: $\quad k \leftarrow \frac{p^p \|\boldsymbol{A}\Delta\|_p^p}{2p^2 \Delta^\top \boldsymbol{A}^\top \boldsymbol{R}\boldsymbol{A}\Delta}$
4: $\quad \alpha_0 \leftarrow \min\left\{ \frac{1}{16\lambda}, \frac{1}{(16\lambda k)^{1/(p-1)}} \right\}$
5: $\quad$ **if** $\gamma(\alpha_0 \cdot \widetilde{\Delta}) < \frac{\alpha_0}{4}i$ or $\Delta^\top \boldsymbol{A}^\top (\boldsymbol{R} + s\boldsymbol{I})\boldsymbol{A}\Delta > \lambda i/p^2$ **then return** true
6: $\quad$ **else return** false
---

Our algorithm $p$-IRLS, described in Algorithm (1), is the standard IRLS algorithm (equation 2) with few key modifications. The first difference is that at each iteration $t$, we add a small systematic padding $s^{(t)}\boldsymbol{I}$ to the weights $\boldsymbol{R}^{(t)}$. The second difference is that the next iterate $\boldsymbol{x}^{(t+1)}$ is calculated by performing a line search along the line joining the current iterate $\boldsymbol{x}^{(t)}$ and the standard IRLS iterate $\widetilde{\boldsymbol{x}}^{(t+1)}$ at iteration $t+1$ (with the modified weights)[2]. Both these modifications have been tried in practice, but primarily from practical justifications: padding the weights avoids ill-conditioned matrices, and line-search can only help us converge faster and improves stability [Kar70, VB99, VB99]. Our key contribution is to show that these modifications together allow us to provably make $\Omega_p(m^{-\frac{p-2}{2(p-1)}})$ progress towards the optimum, resulting in a final iteration count of $O_p(m^{\frac{p-2}{2(p-1)}} \log \frac{m}{\varepsilon})$. Finally, at every iteration we check if the objective value decreases sufficiently, and this allows us to adjust $s^{(t)}$ appropriately. We emphasize here that our algorithm always converges. We prove the following theorem:

**Theorem 3.1.** *Given any $\boldsymbol{A} \in \mathbb{R}^{m \times n}, \boldsymbol{b} \in \mathbb{R}^m, \varepsilon > 0, p \geq 2$ and $\boldsymbol{x}^\star = \arg\min_{\boldsymbol{x}: \boldsymbol{Cx=d}} \|\boldsymbol{Ax} - \boldsymbol{b}\|_p^p$. Algorithm 1 returns $\boldsymbol{x}$ such that $\|\boldsymbol{Ax} - \boldsymbol{b}\|_p^p \leq (1 + \varepsilon) \|\boldsymbol{Ax}^\star - \boldsymbol{b}\|_p^p$ and $\boldsymbol{Cx} = \boldsymbol{d}$, in at most $O\left( p^{3.5} m^{\frac{p-2}{2(p-1)}} \log\left(\frac{m}{\varepsilon}\right) \right)$ iterations.*

The approximation guarantee on the objective value can be translated to a guarantee on coordinate wise convergence. For details on this refer to the supplementary material.

## 3.1 Convergence Analysis

The analysis, at a high level, is based on iterative refinement techniques for $\ell_p$-norms developed in the work of Adil *et al* [AKPS19] and Kyng *et al* [KPSW19]. These techniques allow us to use a crude $\kappa$-approximate solver for the *residual* problem (Definition 2.1) $O_p(\kappa \log \frac{m}{\varepsilon})$ number of times to obtain a $(1 + \varepsilon)$ approximate solution for the $\ell_p$-regression problem (Lemma 3.2).

In our algorithm, if we had solved the standard weighted $\ell_2$ problem instead, $\kappa$ would be unbounded. The padding added to the weights allow us to prove that the solution to weighted $\ell_2$ problem gives a bounded approximation to the residual problem provided we have the correct padding, or in other words correct value of $i$ (Lemma 3.3). We will show that the number of iterations where we are adjusting the value of $i$ are small. Finally, Lemma 3.5 shows that when the algorithm terminates, we have an $\varepsilon$-approximate solution to our main problem. The remaining lemma of this section, Lemma 3.4 gives the loop invariant which is used at several places in the proof of Theorem 3.1. Due to space constraints, we only state the main lemmas here and defer the proofs to the supplementary material.

We begin with the lemma that talks about our overall iterative refinement scheme. The iterative refinement scheme in [AKPS19] and [KPSW19] has an exponential dependence on $p$. We improve this dependence to a small polynomial in $p$.

**Lemma 3.2.** *(Iterative Refinement). Let* $p \geq 2$, *and* $\kappa \geq 1$. *Starting from* $\boldsymbol{x}^{(0)} = \arg\min_{\boldsymbol{Cx}=\boldsymbol{d}} \|\boldsymbol{Ax} - \boldsymbol{b}\|_2^2$, *and iterating as,* $\boldsymbol{x}^{(t+1)} = \boldsymbol{x}^{(t)} - \Delta$, *where* $\Delta$ *is a* $\kappa$-approximate *solution to the residual problem (Definition 2.1), we get an* $\varepsilon$-approximate solution to (3) in at most $O\left(p^2 \kappa \log\left(\frac{m}{\varepsilon}\right)\right)$ *calls to a* $\kappa$-approximate solver for the residual problem.

The next lemma talks about bounding the approximation factor $\kappa$, when we have the right value of $i$.

**Lemma 3.3.** *(Approximation). Let* $\boldsymbol{R}, \boldsymbol{g}, s, \alpha$ *be as defined in lines* (5), (6), (7) *and* (9) *of Algorithm 1. Let* $\alpha_0$ *be as defined in line* (4) *of Algorithm 2 and* $\widetilde{\Delta}$ *be the solution of the following program,*

$$\arg\min_{\Delta} \Delta^\top \boldsymbol{A}^\top (\boldsymbol{R} + s\boldsymbol{I})\boldsymbol{A}\Delta \quad s.t. \quad \boldsymbol{g}^\top \boldsymbol{A}\Delta = i/2, \boldsymbol{C}\Delta = 0. \tag{4}$$

*If* $\widetilde{\Delta}^\top \boldsymbol{A}^\top (\boldsymbol{R} + s\boldsymbol{I})\boldsymbol{A}\widetilde{\Delta} \leq \lambda i/p^2$ *and* $\gamma(\alpha_0 \cdot \widetilde{\Delta}) \geq \frac{\alpha_0 i}{4}$, *then* $\alpha \cdot \widetilde{\Delta}$ *is an* $O\left(p^{1.5} m^{\frac{p-2}{2(p-1)}}\right)$- *approximate solution to the residual problem.*

We next present the loop invariant followed by the conditions for the termination.

**Lemma 3.4.** *(Invariant) At every iteration of the while loop, we have* $\boldsymbol{Cx}^{(t)} = \boldsymbol{d}$, $\frac{(\|\boldsymbol{Ax}^{(t)} - \boldsymbol{b}\|_p^p - \|\boldsymbol{Ax}^\star - \boldsymbol{b}\|_p^p)}{16p} \leq i$ *and* $i \geq \frac{\varepsilon}{16p(1+\varepsilon)} \|\boldsymbol{Ax}^{(0)} - \boldsymbol{b}\|_p^p m^{-(p-2)/2}$.

**Lemma 3.5.** *(Termination). Let* $i$ *be such that* $(\|\boldsymbol{Ax}^{(t)} - \boldsymbol{b}\|_p^p - \|\boldsymbol{Ax}^\star - \boldsymbol{b}\|_p^p)/16p \in (i/2, i]$. *Then,*

$$i \leq \frac{\varepsilon}{16p(1+\varepsilon)} \|\boldsymbol{Ax}^{(t)} - \boldsymbol{b}\|_p^p \Rightarrow \|\boldsymbol{Ax}^{(t)} - \boldsymbol{b}\|_p^p \leq (1+\varepsilon)\text{OPT}.$$

*and,*

$$\|\boldsymbol{Ax}^{(t)} - \boldsymbol{b}\|_p^p \leq (1+\varepsilon)\text{OPT} \Rightarrow i \leq 2\frac{\varepsilon}{16p(1+\varepsilon)} \|\boldsymbol{Ax}^{(t)} - \boldsymbol{b}\|_p^p.$$

We next see how Lemmas 3.2, 3.3, 3.4, and 3.5 together imply our main result, Theorem 3.1.

## 3.2 Proof of Theorem 3.1

*Proof.* We first show that at termination, the algorithm returns an $\varepsilon$-approximate solution. We begin by noting that the quantity $i$ can only decrease with every iteration. At iteration $t$, let $i_0$ denote the smallest number such that $(\|\boldsymbol{Ax}^{(t)} - \boldsymbol{b}\|_p^p - \|\boldsymbol{Ax}^\star - \boldsymbol{b}\|_p^p)/16p \in (i_0/2, i_0]$. Note that $i$ must be at least $i_0$ (Lemma 3.4). Let us first consider the termination condition of the while loop. When we terminate, $\frac{\varepsilon\|\boldsymbol{Ax}^{(t)} - \boldsymbol{b}\|_p^p}{16p(1+\varepsilon)} \geq i \geq i_0$. Lemma 3.5 now implies that $\left\|\boldsymbol{Ax}^{(t)} - \boldsymbol{b}\right\|_p^p \leq (1+\varepsilon)OPT$.

Lemma 3.4 also shows that at each iteration our solution satisfies $\boldsymbol{Cx}^{(t)} = \boldsymbol{d}$, therefore the solution returned at termination also satisfies the subspace constraints.

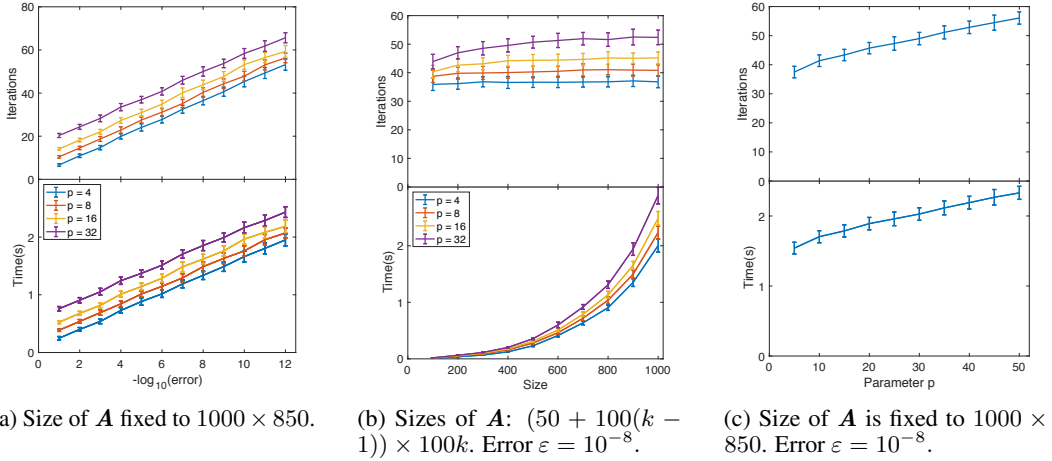

(a) Size of $\boldsymbol{A}$ fixed to $1000 \times 850$.

(b) Sizes of $\boldsymbol{A}$: $(50 + 100(k - 1)) \times 100k$. Error $\varepsilon = 10^{-8}$.

(c) Size of $\boldsymbol{A}$ is fixed to $1000 \times 850$. Error $\varepsilon = 10^{-8}$.

Figure 2: Random Matrix instances. Comparing the number of iterations and time taken by our algorithm with the parameters. Averaged over 100 random samples for $\boldsymbol{A}$ and $\boldsymbol{b}$. Linear solver used : backslash.

We next prove the running time bound. Note that the objective is non increasing with every iteration. This is because the LINESEARCH returns a factor that minimizes the objective given a direction $\widetilde{\Delta}$, i.e., $\alpha = \arg\min_{\delta} \|\boldsymbol{A}(\boldsymbol{x} - \delta\widetilde{\Delta}) - \boldsymbol{b}\|_p^p$, which could also be zero.

We now show that at every iteration the algorithm either reduces $i$ or finds $\widetilde{\Delta}$ that gives a $O\left(p^{1.5} m^{\frac{p-2}{2(p-1)}}\right)$-approximate solution to the residual problem. Consider an iteration where the algorithm does not reduce $i$. It suffices to prove that in this iteration, the algorithm obtains an $O\left(p^{1.5} m^{\frac{p-2}{2(p-1)}}\right)$-approximate solution to the residual problem. Since the algorithm does not reduce $i$, we must have $\gamma(\alpha_0\widetilde{\Delta}) \geq \alpha_0 i/4$, and $\widetilde{\Delta}^{\top} \boldsymbol{A}^{\top}(\boldsymbol{R} + s\boldsymbol{I})\boldsymbol{A}\widetilde{\Delta} \leq \lambda i/p^2$. It follows from Lemma 3.3, we know that $\widetilde{\Delta}$ gives the required approximation to the residual problem.

Thus, the algorithm either reduces $i$ or returns an $O\left(p^{1.5} m^{\frac{p-2}{2(p-1)}}\right)$-approximate solution to the residual problem. The number of steps in which we reduce $i$ is at most $\log(i_{initial}/i_{min}) = p \log\left(\frac{m}{\varepsilon}\right)$ (Lemma 3.4 gives the value of $i_{min}$). By Lemma 3.2, the number of steps where the algorithm finds an approximate solution before it has found a $(1 + \varepsilon)$-approximate solution is at most $O\left(p^{3.5} m^{\frac{p-2}{2(p-1)}} \log\left(\frac{m}{\varepsilon}\right)\right)$. Thus, the total number of iterations required by our algorithm is $O\left(p^{3.5} m^{\frac{p-2}{2(p-1)}} \log\left(\frac{m}{\varepsilon}\right)\right)$, completing the proof of the theorem. $\qquad\square$

## 4 Experiments

In this section, we detail our results from experiments studying the performance of our algorithm, $p$-IRLS. We implemented our algorithm in Matlab on a standard desktop machine, and evaluated its performance on two types of instances, random instances for $\ell_p$-regression, and graphs for $p$-Laplacian minimization. We study the scaling behavior of our algorithm as we change $p, \varepsilon$, and the size of the problem. We compare our performance to the Matlab/CVX solver that is guaranteed to find a good solution, and to the IRLS/homotopy based implementation from [RCL19] that is not guaranteed to converge, but runs quite well in practice. We now describe our instances, parameters and experiments in detail.

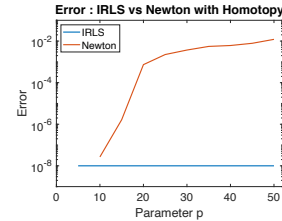

Figure 1: Averaged over 100 random samples. Graph: 1000 nodes (5000-6000 edges). Solver: PCG with Cholesky preconditioner.

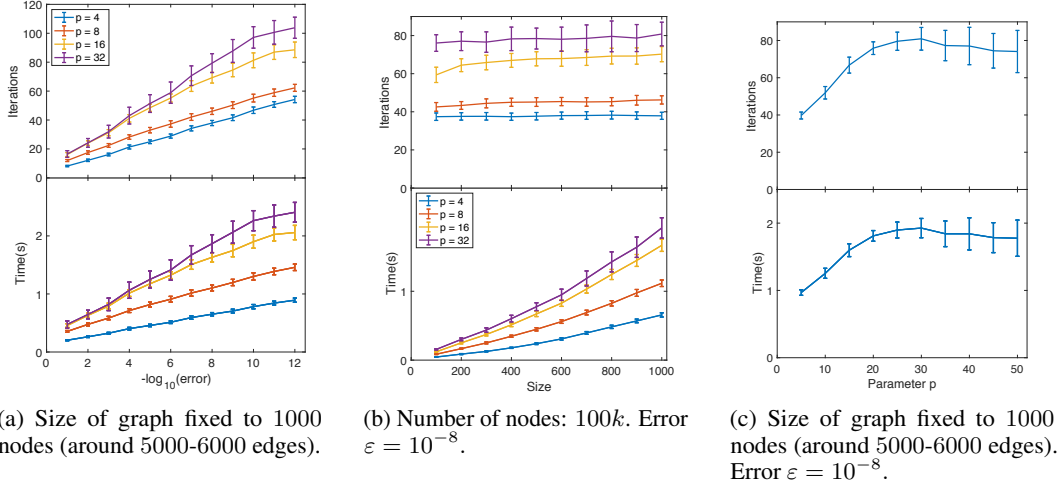

(a) Size of graph fixed to 1000 nodes (around 5000-6000 edges).

(b) Number of nodes: $100k$. Error $\varepsilon = 10^{-8}$.

(c) Size of graph fixed to 1000 nodes (around 5000-6000 edges). Error $\varepsilon = 10^{-8}$.

Figure 3: Graph Instances. Comparing the number of iterations and time taken by our algorithm with the parameters. Averaged over 100 graph samples. Linear solver used : backslash.

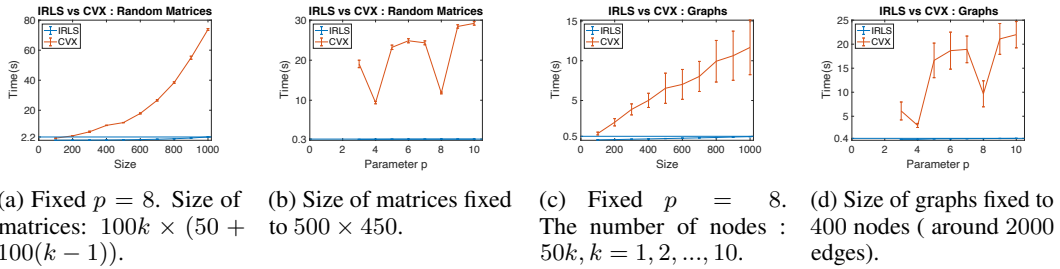

(a) Fixed $p = 8$. Size of matrices: $100k \times (50 + 100(k-1))$.

(b) Size of matrices fixed to $500 \times 450$.

(c) Fixed $p = 8$. The number of nodes : $50k, k = 1, 2, ..., 10$.

(d) Size of graphs fixed to 400 nodes ( around 2000 edges).

Figure 4: Averaged over 100 samples. Precision set to $\varepsilon = 10^{-8}$. CVX solver used : SDPT3 for Matrices and Sedumi for Graphs.

**Instances and Parameters.** We consider two types of instances, random matrices and graphs.

1. **Random Matrices:** We want to solve the problem $\min_x \| Ax - b \|_p$. In these instances we use random matrices $A$ and $b$, where every entry of the matrix is chosen uniformly at random between $0$ and $1$.

2. **Graphs:** We use the graphs described in [RCL19]. The set of vertices is generated by choosing vectors in $[0,1]^{10}$ uniformly at random and the edges are created by connecting the 10 nearest neighbours. Weights of each edge is specified by a gaussian type function (Eq 3.1,[RCL19]). Very few vertices (around 10) have labels which are again chosen uniformly at random between $0$ and $1$. The problem studied on these instances is to determine the minimizer of the $\ell_p$ laplacian. We formulate this problem into the form $\min_x \| Ax - b \|_p^p$, details of this formulation can be found in the Appendix that is in the supplementary material.

Note that we have 3 different parameters for each problem, the size of the instance i.e., the number of rows of matrix $A$, the norm we solve for, $p$, and the accuracy to which we want to solve each problem, $\varepsilon$. We will consider each of these parameters independently and see how our algorithm scales with them for both instances.

**Benchmark Comparisons.** We compare the performance of our program with the following:

1. Standard MATLAB optimization package, CVX [GB14, GB08].

2. The most efficient algorithm for $\ell_p$-semi supervised learning given in [RCL19] was newton's method with homotopy. We take their hardest problem, and compare the performance of their code with ours by running our algorithm for the same number of iterations as them and showing that we get closer to the optimum, or in other words a smaller error $\varepsilon$, thus showing we converge much faster.

**Implementation Details.**    We normalize the instances by running our algorithm once and dividing the vector $b$ by the norm of the final objective, so that our norms at the end are around 1. We do this for every instance before we measure the runtime or the iteration count for uniformity and to avoid numerical precision issues. All experiments were performed on MATLAB 2018b on a Desktop ubuntu machine with an Intel Core $i5$-4570 CPU @ $3.20GHz \times 4$ processor and 4GB RAM. For the graph instances, we fix the dimension of the space from which we choose vertices to 10 and the number of labelled vertices to be 10. The graph instances are generated using the code [Rio19] by [RCL19]. Other details specific to the experiment are given in the captions.

## 4.1    Experimental Results

**Dependence on Parameters.**    Figure 2 shows the dependence of the number of iterations and runtime on our parameters for random matrices. Similarly for graph instances, Figure 3 shows the dependence of iteration count and runtime with the parameters. As expected from the theoretical guarantees, the number of iterations and runtimes increase linearly with $\log\left(\frac{1}{\varepsilon}\right)$. The dependence on size and $p$ are clearly much better in practice (nearly constant and at most linear respectively) than the theoretical bounds ($m^{1/2}$ and $p^{3.5}$ respectively) for both kinds of instances.

**Comparisons with Benchmarks.**

- Figure 4 shows the runtime comparison between our IRLS algorithm $p$-IRLS and CVX. For all instances, we ensured that our final objective was smaller than the objective of the CVX solver. As it is clear for both kinds of instances, our algorithm takes a lot lesser time and also increases more slowly with size and $p$ as compared to CVX. Note that that CVX does a lot better when $p = 2^k$, but it is still at least 30-50 times slower for random matrices and 10-30 times slower for graphs.
- Figure 1 shows the performance of our algorithm when compared to the IRLS/Homotopy method of [RCL19]. We use the same linear solvers for both programs, preconditioned conjugate gradient with an incomplete cholesky preconditioner and run both programs to the same number of iterations. The plots indicate the value $\varepsilon$ as described previously. For our IRLS algorithm we indicate our upper bound on $\varepsilon$ and for their procedure we indicate a lower bound on $\varepsilon$ which is the relative difference in the objectives achieved by the two algorithms. It is clear that our algorithm achieves an error that is orders of magnitudes smaller than the error achieved by their algorithm. This shows that our algorithm has a much faster rate of convergence. Note that there is no guarantee on the convergence of the method used by [RCL19], whereas we prove that our algorithm converges in a small number of iterations.

## 5    Discussion

To conclude, we present $p$-IRLS, the first IRLS algorithm that provably converges to a high accuracy solution in a small number of iterations. This settles a problem that has been open for over three decades. Our algorithm is very easy to implement and we demonstrate that it works very well in practice, beating the standard optimization packages by large margins. The theoretical bound on the numbers of iterations has a sub-linear dependence on size and a small polynomial dependence on $p$, however in practice, we see an almost constant dependence on size and at most linear dependence on $p$ in random instances and graphs. In order to achieve the best theoretical bounds we would require some form of acceleration. For $\ell_1$ and $\ell_\infty$ regression, it has been shown that it is possible to achieve acceleration, however without geometric convergence. It remains an open problem to give a practical IRLS algorithm which simultaneously has the best possible theoretical convergence bounds.

## Acknowledgements

DA is supported by SS's NSERC Discovery grant and an Ontario Graduate Scholarship. SS is supported by the Natural Sciences and Engineering Research Council of Canada (NSERC), a Connaught New Researcher award, and a Google Faculty Research award. RP is partially supported by the NSF under Grants No. 1637566 and No. 1718533.

## Footnotes

[2]Note that $p$-IRLS has been written in a slightly different but equivalent formulation, where it solves for $\widetilde{\Delta} = \boldsymbol{x}^{(t)} - \widetilde{\boldsymbol{x}}^{(t+1)}$.

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
