[Supplementary Material]

## Appendix A   Coordinate Wise Convergence vs Convergence in Objective Value

For an algorithm with a $\log \frac{1}{\varepsilon}$ dependence of the running time for computing a $(1+\varepsilon)$-approximate solution, like $p$-IRLS, the guarantee can be translated into a guarantee for convergence in the solution without any significant loss in the runtime complexity of the method. We demonstrate this theoretically and experimentally below.

**Lemma A.1.** *If $\boldsymbol{x}$ is a $(1+\delta)$-approximate solution and $\boldsymbol{x}^\star$ is the optimum, then*

$$\|\boldsymbol{x} - \boldsymbol{x}^\star\|_\infty \leq \frac{2m^{\frac{1}{2}}}{\sigma_{\min}(\boldsymbol{A})} \left(\frac{2\delta}{m}\right)^{\frac{1}{p}} \|\boldsymbol{A}\boldsymbol{x}^\star - \boldsymbol{b}\|_p \,,$$

*where $\sigma_{\min}(\boldsymbol{A})$ is the smallest singular value of $\boldsymbol{A}$.*

*Proof.* Given that $\boldsymbol{x}$ is a $(1+\delta)$-approximate solution, using Lemma B.1, we can write the following lower bound on the objective value:

$$(1+\delta)\|\boldsymbol{A}\boldsymbol{x}^\star - \boldsymbol{b}\|_p^p \geq \|\boldsymbol{A}\boldsymbol{x}^\star - \boldsymbol{b}\|_p^p + p\left(\boldsymbol{A}\boldsymbol{x}^\star - \boldsymbol{b}\right)^\top \boldsymbol{R}\boldsymbol{A}(\boldsymbol{x} - \boldsymbol{x}^\star)$$
$$+ \, {p}/{8}\boldsymbol{A}(\boldsymbol{x} - \boldsymbol{x}^\star)^\top \boldsymbol{A}^\top \boldsymbol{R}\boldsymbol{A}(\boldsymbol{x} - \boldsymbol{x})^\star + 2^{-(p+1)}\|\boldsymbol{A}\boldsymbol{x} - \boldsymbol{A}\boldsymbol{x}^\star\|_p^p \,,$$

where $\boldsymbol{R} = \mathrm{diag}(|\boldsymbol{A}\boldsymbol{x}^\star - \boldsymbol{b}|^{p-2})$. Since the gradient at $\boldsymbol{x}^\star$ is $0$, simplifying, we get, $2^{p+1}\delta\|\boldsymbol{A}\boldsymbol{x}^\star - \boldsymbol{b}\|_p^p \geq \|\boldsymbol{A}\boldsymbol{x} - \boldsymbol{A}\boldsymbol{x}^\star\|_p^p$. Now, translating between various norms, we obtain,

$$\|\boldsymbol{x} - \boldsymbol{x}^\star\|_\infty \leq \frac{1}{\sigma_{\min}(\boldsymbol{A})}\|\boldsymbol{A}\boldsymbol{x} - \boldsymbol{A}\boldsymbol{x}^\star\|_2 \leq \frac{m^{\frac{1}{2}-\frac{1}{p}}}{\sigma_{\min}(\boldsymbol{A})}\|\boldsymbol{A}\boldsymbol{x} - \boldsymbol{A}\boldsymbol{x}^\star\|_p \leq \frac{2m^{\frac{1}{2}}}{\sigma_{\min}(\boldsymbol{A})}\left(\frac{2\delta}{m}\right)^{\frac{1}{p}}\|\boldsymbol{A}\boldsymbol{x}^\star - \boldsymbol{b}\|_p \,.$$

$\square$

We can achieve the guarantee $\|\boldsymbol{x} - \boldsymbol{x}^\star\|_\infty \leq \varepsilon \|\boldsymbol{A}\boldsymbol{x}^\star - \boldsymbol{b}\|_p$ by picking $\delta = \left(\frac{\varepsilon\sigma_{\min}(\boldsymbol{A})}{4m}\right)^p$. This gives $\log\frac{m}{\delta} = O(p\log\frac{m}{\sigma_{\min}(\boldsymbol{A})\varepsilon})$, and hence a total iteration count of $O(p^{4.5}m^{\frac{p-2}{2(p-1)}}\log\frac{m}{\sigma_{\min}(\boldsymbol{A})\varepsilon})$. Asymptotically, the running time bound is only off by a factor of $p$ if we wish to measure the convergence in $\ell_\infty$-norm, as long as $\log\frac{1}{\sigma_{\min}(\boldsymbol{A})} = O(\log\frac{m}{\varepsilon})$.

Figure 5: Maximum coordinate wise difference with the optimum vs accuracy to which the objective is close to the optimum, for both graphs and random matrix instances.

We also demonstrate this relation experimentally. The plots in Figure 5 demonstrate the average resulting $\ell_\infty$ norm deviation for the solution computed, as we change the $\varepsilon$ parameter used in the algorithm. We use the instances described in the paper; matrices of size $1000 \times 800$ and graphs with $1000$ nodes. For each instance, we: 1) find a very high accuracy solution, by choosing a very small $\varepsilon \sim 10^{-25}$, 2) scale the problem so that the optimum value is $1$, and run the algorithm again to find the optimum solution $\boldsymbol{x}^\star$. 3) Now we have a problem such that $\|\boldsymbol{A}\boldsymbol{x}^\star - \boldsymbol{b}\|_p = 1$, we run the

algorithm again with various values of $\varepsilon$, to obtain solutions $\boldsymbol{x}(\varepsilon)$ and plot $\left\|\boldsymbol{x}(\varepsilon) - \boldsymbol{x}^\star\right\|_\infty$ (averaged over 20 samples). These results are very much in agreement with the theoretical $\varepsilon^{\frac{1}{p}}$ dependence proved above. (Note that the error bars indicate $\log(\text{mean} \pm \text{std})$ so they are missing on one side when mean $<$ std.)

## Appendix B   Proofs from Section 3

### B.1   Proof of Lemma 3.2

**Lemma 3.2.** *(Iterative Refinement).*     *Let* $p \geq 2$, *and* $\kappa \geq 1$.   *Starting from* $\boldsymbol{x}^{(0)} = \arg\min_{\boldsymbol{C}\boldsymbol{x}=\boldsymbol{d}} \left\|\boldsymbol{A}\boldsymbol{x} - \boldsymbol{b}\right\|_2^2$, *and iterating as,* $\boldsymbol{x}^{(t+1)} = \boldsymbol{x}^{(t)} - \Delta$, *where* $\Delta$ *is a* $\kappa$-*approximate solution to the residual problem (Definition 2.1), we get an* $\varepsilon$-*approximate solution to* (3) *in at most* $O\left(p^2 \kappa \log\left(\frac{m}{\varepsilon}\right)\right)$ *calls to a* $\kappa$-*approximate solver for the residual problem.*

We first show that we can upper and lower bound the change in objective by a linear term plus a quadratically smoothed function.

**Lemma B.1.** *For any* $\boldsymbol{x}, \Delta$ *and* $p \geq 2$, *we have for* $\boldsymbol{r} = |\boldsymbol{x}|^{p-2}$ *and* $\boldsymbol{g} = p|\boldsymbol{x}|^{p-2}\boldsymbol{x}$,

$$\frac{p}{8}\sum_e \boldsymbol{r}_e \Delta_e^2 + \frac{1}{2^{p+1}}\|\Delta\|_p^p \leq \|\boldsymbol{x} + \Delta\|_p^p - \|\boldsymbol{x}\|_p^p - \boldsymbol{g}^\top \Delta \leq 2p^2 \sum_e \boldsymbol{r}_e \Delta_e^2 + p^p \|\Delta\|_p^p.$$

The proof of the above lemma is long and hence deferred to the end of this section. Applying the above lemma on our objective we get,

$$\frac{p}{8}(\boldsymbol{A}\Delta)^\top \boldsymbol{R}\boldsymbol{A}\Delta + \frac{1}{2^{p+1}}\|\boldsymbol{A}\Delta\|_p^p \leq \left\|\boldsymbol{A}(\boldsymbol{x} + \Delta) - \boldsymbol{b}\right\|_p^p - \|\boldsymbol{A}\boldsymbol{x} - \boldsymbol{b}\|_p^p - \boldsymbol{g}^\top \boldsymbol{A}\Delta \leq 2p^2(\boldsymbol{A}\Delta)^\top \boldsymbol{R}\boldsymbol{A}\Delta + p^p \|\boldsymbol{A}\Delta\|_p^p,$$
$$(5)$$

where $\boldsymbol{R}$ is the diagonal matrix with entries $|\boldsymbol{A}\boldsymbol{x} - \boldsymbol{b}|^{p-2}$ and $\boldsymbol{g} = p\boldsymbol{R}(\boldsymbol{A}\boldsymbol{x} - \boldsymbol{b})$. We next show the relation between the residual problem defined in the preliminaries and the change in objective value when $\boldsymbol{x}$ is updated by $\Delta$.

**Lemma B.2.** *For any* $\boldsymbol{x}, \Delta$ *and* $p \geq 2$ *and* $\lambda = 16p$,

$$\gamma(\Delta) \leq \|\boldsymbol{A}\boldsymbol{x} - \boldsymbol{b}\|_p^p - \left\|\boldsymbol{A}(\boldsymbol{x} - \Delta) - \boldsymbol{b}\right\|_p^p,$$

*and*

$$\|\boldsymbol{A}\boldsymbol{x} - \boldsymbol{b}\|_p^p - \left\|\boldsymbol{A}(\boldsymbol{x} - \lambda\Delta) - \boldsymbol{b}\right\|_p^p \leq \lambda\gamma(\Delta).$$

*Proof.* The first inequality directly follows from (5). For the second inequality,

$$\|\boldsymbol{A}\boldsymbol{x} - \boldsymbol{b}\|_p^p - \left\|\boldsymbol{A}(\boldsymbol{x} - \lambda\Delta) - \boldsymbol{b}\right\|_p^p \leq \lambda \boldsymbol{g}^\top \Delta - \lambda^2 \frac{p}{8}\Delta^\top \boldsymbol{A}^\top \boldsymbol{R}\boldsymbol{A}\Delta - \lambda^p \frac{1}{2^{p+1}}\|\boldsymbol{A}\Delta\|_p^p$$

$$= \lambda\left(\boldsymbol{g}^\top \Delta - \lambda\frac{p}{8}\Delta^\top \boldsymbol{A}^\top \boldsymbol{R}\boldsymbol{A}\Delta - \lambda^{p-1}\frac{1}{2^{p+1}}\|\boldsymbol{A}\Delta\|_p^p\right)$$

$$\leq \lambda\left(\boldsymbol{g}^\top \boldsymbol{A}\Delta - 2p^2 \Delta^\top \boldsymbol{A}^\top \boldsymbol{R}\boldsymbol{A}\Delta - p^p \|\boldsymbol{A}\Delta\|_p^p\right).$$

$$\square$$

### B.1.1   Proof of Lemma Iterative Refinement

*Proof.* Let $\widetilde{\Delta}$ be a $\kappa$-approximate solution to the residual problem. Using this fact and Lemma B.2 for $\Delta = \frac{\boldsymbol{x} - \boldsymbol{x}^\star}{\lambda}$, we get,

$$\gamma(\widetilde{\Delta}) \geq \frac{1}{\kappa}\gamma(\Delta^\star) \geq \frac{1}{\kappa}\gamma\left(\frac{\boldsymbol{x} - \boldsymbol{x}^\star}{\lambda}\right) \geq \frac{1}{\lambda\kappa}\left(\|\boldsymbol{A}\boldsymbol{x} - \boldsymbol{b}\|_p^p - OPT\right).$$

Also,

$$\left\|A(x-\widetilde\Delta)-b\right\|_p^p - OPT \le \|Ax-b\|_p^p - \gamma(\widetilde\Delta) - OPT$$

$$\le \left(\|Ax-b\|_p^p - OPT\right) - \frac{1}{\lambda\kappa}\left(\|Ax-b\|_p^p - OPT\right)$$

$$= \left(1-\frac{1}{\lambda\kappa}\right)\left(\|Ax-b\|_p^p - OPT\right).$$

Now, after $t$ iterations,

$$\left\|A(x^{(t)}-\widetilde\Delta)-b\right\|_p^p - OPT \le \left(1-\frac{1}{\lambda\kappa}\right)^t\left(\left\|Ax^{(0)}-b\right\|_p^p - OPT\right) \le \left(1-\frac{1}{\lambda\kappa}\right)^t m^{(p-2)/2}OPT.$$

Thus, for our value of $\lambda = 16p$, $8p^2\kappa\log(m/\varepsilon)$ iterations suffice to obtain a $(1+\varepsilon)$-approximate solution. $\qquad\square$

### B.1.2 Proof of Lemma B.1

*Proof.* To show this, we show that the above holds for all coordinates. For a single coordinate, the above expression is equivalent to proving,

$$\frac{p}{8}|x|^{p-2}\Delta^2 + \frac{1}{2^{p+1}}|\Delta|^p \le |x+\Delta|^p - |x|^p - p|x|^{p-1}\,sgn(x)\Delta \le 2p^2|x|^{p-2}\Delta^2 + p^p|\Delta|^p.$$

Let $\Delta = \alpha x$. Since the above clearly holds for $x=0$, it remains to show for all $\alpha$,

$$\frac{p}{8}\alpha^2 + \frac{1}{2^{p+1}}|\alpha|^p \le |1+\alpha|^p - 1 - p\alpha \le 2p^2\alpha^2 + p^p|\alpha|^p.$$

1. $\alpha \ge 1$:
   In this case, $1+\alpha \le 2\alpha \le p\cdot\alpha$. So, $|1+\alpha|^p \le p^p|\alpha|^p$ and the right inequality directly holds. To show the other side, let

   $$h(\alpha) = (1+\alpha)^p - 1 - p\alpha - \frac{p}{8}\alpha^2 - \frac{1}{2^{p+1}}\alpha^p.$$

   We have,

   $$h'(\alpha) = p(1+\alpha)^{p-1} - p - \frac{p}{4}\alpha - \frac{p}{2^{p+1}}\alpha^{p-1}$$

   and

   $$h''(\alpha) = p(p-1)(1+\alpha)^{p-2} - \frac{p}{4} - \frac{p(p-1)}{2^{p+1}}\alpha^{p-2} \ge 0.$$

   Since $h''(\alpha) \ge 0$, $h'(\alpha) \ge h'(1) \ge 0$. So $h$ is an increasing function in $\alpha$ and $h(\alpha) \ge h(1) \ge 0$.

2. $\alpha \le -1$:
   Now, $|1+\alpha| \le 1+|\alpha| \le p\cdot|\alpha|$, and $2\alpha^2p^2 - |\alpha|\,p \ge 0$. As a result,

   $$|1+\alpha|^p \le -|\alpha|\,p + 2\alpha^2p^2 + p^p\cdot|\alpha|^p$$

   which gives the right inequality. Consider,

   $$h(\alpha) = |1+\alpha|^p - 1 - p\alpha - \frac{p}{8}\alpha^2 - \frac{1}{2^{p+1}}|\alpha|^p.$$

   $$h'(\alpha) = -p|1+\alpha|^{p-1} - p - \frac{p}{4}\alpha + p\frac{1}{2^{p+1}}|\alpha|^{p-1}.$$

   Let $\beta = -\alpha$. The above expression now becomes,

   $$-p(\beta-1)^{p-1} - p + \frac{p}{4}\beta + p\frac{1}{2^{p+1}}\beta^{p-1}.$$

   We know that $\beta \ge 1$. When $\beta \ge 2$, $\frac{\beta}{2} \le \beta-1$ and $\frac{\beta}{2} \le \left(\frac{\beta}{2}\right)^{p-1}$. This gives us,

   $$\frac{p}{4}\beta + p\frac{1}{2^{p+1}}\beta^{p-1} \le \frac{p}{2}\left(\frac{\beta}{2}\right)^{p-1} + \frac{p}{2}\left(\frac{\beta}{2}\right)^{p-1} \le p(\beta-1)^{p-1}$$

giving us $h'(\alpha) \le 0$ for $\alpha \le -2$. When $\beta \le 2$, $\frac{\beta}{2} \ge \left(\frac{\beta}{2}\right)^{p-1}$ and $\frac{\beta}{2} \le 1$.

$$\frac{p}{4}\beta + p\frac{1}{2^{p+1}}\beta^{p-1} \le \frac{p}{2} \cdot \frac{\beta}{2} + \frac{p}{2} \cdot \frac{\beta}{2} \le p$$

giving us $h'(\alpha) \le 0$ for $-2 \le \alpha \le -1$. Therefore, $h'(\alpha) \le 0$ giving us, $h(\alpha) \ge h(-1) \ge 0$, thus giving the left inequality.

3. $|\alpha| \le 1$:
Let $s(\alpha) = 1 + p\alpha + 2p^2\alpha^2 + p^p |\alpha|^p - (1+\alpha)^p$. Now,

$$s'(\alpha) = p + 4p^2\alpha + p^{p+1} |\alpha|^{p-1} sgn(\alpha) - p(1+\alpha)^{p-1}.$$

When $\alpha \le 0$, we have,

$$s'(\alpha) = p + 4p^2\alpha - p^{p+1} |\alpha|^{p-1} - p(1+\alpha)^{p-1}.$$

and

$$s''(\alpha) = 4p^2 + p^{p+1}(p-1) |\alpha|^{p-2} - p(p-1)(1+\alpha)^{p-1} \ge 2p^2 + p^{p+1}(p-1) |\alpha|^{p-2} - p(p-1) \ge 0.$$

So $s'$ is an increasing function of $\alpha$ which gives us, $s'(\alpha) \le s'(0) = 0$. Therefore $s$ is a decreasing function, and the minimum is at $0$ which is $0$. This gives us our required inequality for $\alpha \le 0$. When $\alpha \ge \frac{1}{p-1}$, $1 + \alpha \le p \cdot \alpha$ and $s'(\alpha) \ge 0$. We are left with the range $0 \le \alpha \le \frac{1}{p-1}$. Again, we have,

$$s''(\alpha) = 4p^2 + p^{p+1}(p - 1) |\alpha|^{p-2} - p(p - 1)(1 + \alpha)^{p-1}$$

$$\ge 4p^2 + p^{p+1}(p - 1) |\alpha|^{p-2} - p(p - 1)(1 + \frac{1}{p - 1})^{p-1}$$

$$\ge 4p^2 + p^{p+1}(p - 1) |\alpha|^{p-2} - p(p - 1)e, \text{ When } p \text{ gets large the last term approaches } e$$

$$\ge 0.$$

Therefore, $s'$ is an increasing function, $s'(\alpha) \ge s'(0) = 0$. This implies $s$ is an increasing function, giving, $s(\alpha) \ge s(0) = 0$ as required.

To show the other direction,

$$h(\alpha) = (1+\alpha)^p - 1 - p\alpha - \frac{p}{8}\alpha^2 - \frac{1}{2^{p+1}} |\alpha|^p \ge (1+\alpha)^p - 1 - p\alpha - \frac{p}{8}\alpha^2 - \frac{p}{8}\alpha^2 = (1+\alpha)^p - 1 - p\alpha - \frac{p}{4}\alpha^2.$$

Now, since $p \ge 2$,

$$\left((1 + \alpha)^{p-2} - 1\right) sgn(\alpha) \ge 0$$

$$\Rightarrow \left((1 + \alpha)^{p-1} - 1 - \alpha\right) sgn(\alpha) \ge 0$$

$$\Rightarrow \left(p(1 + \alpha)^{p-1} - p - \frac{p}{2}\alpha\right) sgn(\alpha) \ge 0$$

We thus have, $h'(\alpha) \ge 0$ when $\alpha$ is positive and $h'(\alpha) \le 0$ when $\alpha$ is negative. The minimum of $h$ is at $0$ which is $0$. This concludes the proof of this case.

$\square$

## B.2 Proof of Lemma that Checks Progress in Objective

We will next prove the following Lemma which shows that we do not change $i$ when we have the correct value of $i$.

**Lemma B.3.** *(Check Progress). Let $\alpha_0$ be as defined in line (4) of Algorithm 2 and $\widetilde{\Delta}$ the solution of program (4). If $i/2 < \frac{(\|Ax^{(t)} - b\|_p^p - \|Ax^\star - b\|_p^p)}{16p} \le i$, then $\gamma(\alpha_0 \cdot \widetilde{\Delta}) \ge \frac{\alpha_0 i}{4}$ and $(A\widetilde{\Delta})^\top(R + sI)A\widetilde{\Delta} \le \lambda i/p^2$.*

We require bounding the objective of program 4. To do that we first give a bound on a decision version of the residual problem, and then relate this problem with problem 4.

**Lemma B.4.** *Let $i$ be such that the optimum of the residual problem, $\gamma(\Delta^\star) \in (i/2, \lambda i]$. Then the following problem has optimum at most $\lambda i$.*

$$\min_{\Delta \in \mathbb{R}^m} \quad 2p^2 (\boldsymbol{A}\Delta)^\top \boldsymbol{R} \boldsymbol{A}\Delta + p^p \|\boldsymbol{A}\Delta\|_p^p$$
$$\boldsymbol{g}^\top \boldsymbol{A}\Delta = i/2 \tag{6}$$
$$\boldsymbol{C}\Delta = 0.$$

*Proof.* The assumption on the residual is

$$\gamma(\Delta^\star) = \boldsymbol{g}^\top \boldsymbol{A}\Delta^\star - 2p^2 (\boldsymbol{A}\Delta^\star)^\top \boldsymbol{R} \boldsymbol{A}\Delta^\star - p^p \|\boldsymbol{A}\Delta^\star\|_p^p \in (i/2, \lambda i].$$

Since the last 2 terms are strictly non-positive, we must have, $\boldsymbol{g}^\top \boldsymbol{A}\Delta^\star \geq i/2$. Since $\Delta^\star$ is the optimum and satisfies $\boldsymbol{C}\Delta^\star = 0$,

$$\frac{d}{d\lambda}\left(\boldsymbol{g}^\top \lambda \boldsymbol{A}\Delta^\star - 2p^2 \lambda^2 (\boldsymbol{A}\Delta^\star)^\top \boldsymbol{R} \boldsymbol{A}\Delta^\star - \lambda^p p^p \|\boldsymbol{A}\Delta^\star\|_p^p\right)_{\lambda=1} = 0.$$

Thus,

$$\boldsymbol{g}^\top \boldsymbol{A}\Delta^\star - 2p^2 (\boldsymbol{A}\Delta^\star)^\top \boldsymbol{R} \boldsymbol{A}\Delta^\star - p^p \|\boldsymbol{A}\Delta^\star\|_p^p = 2p^2 (\boldsymbol{A}\Delta^\star)^\top \boldsymbol{R} \boldsymbol{A}\Delta^\star + (p-1)p^p \|\boldsymbol{A}\Delta^\star\|_p^p.$$

Since $p \geq 2$, we get the following

$$2p^2 (\boldsymbol{A}\Delta^\star)^\top \boldsymbol{R} \boldsymbol{A}\Delta^\star + p^p \|\boldsymbol{A}\Delta^\star\|_p^p \leq \boldsymbol{g}^\top \boldsymbol{A}\Delta^\star - 2p^2 (\boldsymbol{A}\Delta^\star)^\top \boldsymbol{R} \boldsymbol{A}\Delta^\star - p^p \|\boldsymbol{A}\Delta^\star\|_p^p \leq \lambda i.$$

For notational convenience, let function $h_p(\boldsymbol{r}, \Delta) = 2p^2 (\boldsymbol{A}\Delta)^\top \boldsymbol{R} \boldsymbol{A}\Delta + p^p \|\boldsymbol{A}\Delta\|_p^p$. Now, we know that, $\boldsymbol{g}^\top \boldsymbol{A}\Delta^\star \geq i/2$ and $\boldsymbol{g}^\top \boldsymbol{A}\Delta^\star - h_p(\boldsymbol{r}, \Delta^\star) \leq \lambda i$. This gives,

$$i/2 \leq \boldsymbol{g}^\top \boldsymbol{A}\Delta^\star \leq h_p(\boldsymbol{r}, \Delta^\star) + \lambda i \leq 2\lambda i.$$

Let $\Delta = \delta \Delta^\star$, where $\delta = \frac{i}{2\boldsymbol{g}^\top \boldsymbol{A}\Delta^\star}$. Note that $\delta \in [1/4\lambda, 1]$. Now, $\boldsymbol{g}^\top \boldsymbol{A}\Delta = i/2$ and,

$$h_p(\boldsymbol{r}, \Delta) \leq \max\{\delta^2, \delta^p\} h_p(\boldsymbol{r}, \Delta^\star) \leq \lambda i.$$

Note that this $\Delta$ satisfies the constraints of program (6) and has an optimum at most $\lambda i$. So the optimum of the program must have an objective at most $\lambda i$. $\square$

**Claim B.5.** *If the optimal objective of program (6) is at most $Z$, then the optimum objective of program (4) is at most $\frac{Z}{2p^2} + \frac{i^{(p-2)/p} Z^{2/p}}{2p^2}$.*

*Proof.* Let $\Delta^\star$ denote the optimizer of (6) and $\widetilde{\Delta}$ be the optimizer of (4). Since the optimum objective of (6) is at most $Z$, we have we have $\|\boldsymbol{A}\Delta^\star\|_p^p \leq \frac{Z}{p^p}$. This implies that $\Delta^{\star\top} \boldsymbol{A}^\top \boldsymbol{A}\Delta^\star \leq \frac{Z^{2/p}}{p^2} m^{(p-2)/p}$. Since $\Delta^\star$ is a feasible solution of (4), we have for our value of $s$,

$$\widetilde{\Delta}^\top \boldsymbol{A}^\top (\boldsymbol{R} + s^{(t)} \boldsymbol{I}) \boldsymbol{A}\widetilde{\Delta} \leq \Delta^{\star\top} \boldsymbol{A}^\top (\boldsymbol{R} + s^{(t)} \boldsymbol{I}) \boldsymbol{A}\Delta^\star \leq \frac{Z}{2p^2} + \frac{i^{(p-2)/p} Z^{2/p}}{2p^2}.$$

$\square$

**Proof of Lemma B.3**

*Proof.* Since,

$$i/2 < \frac{(\|\boldsymbol{A}\boldsymbol{x}^{(t)} - \boldsymbol{b}\|_p^p - \|\boldsymbol{A}\boldsymbol{x}^\star - \boldsymbol{b}\|_p^p)}{16p} \leq i,$$

we know that the optimum of the residual problem lies between $(i/2, \lambda i]$ (Lemma B.2). From Lemma B.4 the optimum of the problem (6) is at most $\lambda i$. Now, from Claim B.5, we know that $(A\widetilde{\Delta})^\top (R + sI)(A\widetilde{\Delta}) \leq \lambda i/p^2$. Also, note that $\alpha_0 + k\alpha_0^{p-1} \leq \frac{1}{8\lambda}$. Consider the following,

$$
\begin{aligned}
\gamma(\alpha_0 \cdot \widetilde{\Delta}) =\ & \alpha_0 g^\top A\widetilde{\Delta} - \alpha_0^2 2p^2 (A\widetilde{\Delta})^\top R(A\widetilde{\Delta}) - \alpha_0^p p^p \left\| A\widetilde{\Delta} \right\|_p^p \\
\geq\ & \alpha_0 g^\top A\widetilde{\Delta} - (\alpha_0^2 + k\alpha_0^p)2p^2 (A\widetilde{\Delta})^\top (R + sI)(A\widetilde{\Delta}) \\
=\ & \alpha_0 \left( g^\top A\widetilde{\Delta} - (\alpha_0 + k\alpha_0^{p-1})2p^2 (A\widetilde{\Delta})^\top (R + sI)(A\widetilde{\Delta}) \right) \\
\geq\ & \alpha_0 \left( g^\top A\widetilde{\Delta} - \frac{1}{8\lambda}2p^2 (A\widetilde{\Delta})^\top (R + sI)(A\widetilde{\Delta}) \right) \\
\geq\ & \alpha_0 \left( \frac{i}{2} - \frac{1}{8\lambda}2\lambda i \right) \\
\geq\ & \frac{\alpha_0}{4} i
\end{aligned}
$$

$\square$

## B.3 Proof of Lemma 3.4

**Lemma 3.4.** *(Invariant)* *At every iteration of the while loop, we have* $Cx^{(t)} = d$, $\frac{(\|Ax^{(t)} - b\|_p^p - \|Ax^\star - b\|_p^p)}{16p} \leq i$ *and* $i \geq \frac{\varepsilon}{16p(1+\varepsilon)}\|Ax^{(0)} - b\|_p^p m^{-(p-2)/2}$.

*Proof.* We use induction to show this. Initially we set, $i = \|Ax^{(0)} - b\|_p^p/16p$. When the optimum is not 0, this is greater than $(\|Ax^{(0)} - b\|_p^p - \|Ax^\star - b\|_p^p)/16p$. When the optimum is 0, the initial solution (2-norm minimizer) will also give zero and we can stop our procedure. Therefore, the claim holds for $t = 1$. Suppose at iteration $t$ the claim holds. Since the objective is non-increasing, we know that,

$$\|Ax^{(t)} - b\|_p^p - \|Ax^\star - b\|_p^p \geq \|Ax^{(t+1)} - b\|_p^p - \|Ax^\star - b\|_p^p.$$

Let $\widetilde{\Delta}$ denote the solution returned in iteration $t+1$. At iteration $t+1$, if $(\|Ax^{(t+1)} - b\|_p^p - \|Ax^\star - b\|_p^p)/16p \in (i/2, i]$, from Lemma B.3, we will always have $\gamma(\alpha_0\widetilde{\Delta}) \geq \alpha_0 i/4$ and $(A\widetilde{\Delta})^\top (R + sI)A\widetilde{\Delta} \leq \lambda i/p^2$. So the algorithm does not reduce $i$ and as a result our claim holds for $t+1$. Otherwise, we know that $(\|Ax^{(t+1)} - b\|_p^p - \|Ax^\star - b\|_p^p)/16p \leq i/2$ and the algorithm might reduce $i$ by half if either of the two conditions are true. However, the claim still holds. Therefore, $i$ is always at least $(\|Ax^{(t+1)} - b\|_p^p - \|Ax^\star - b\|_p^p)/16p$.

We start with a solution $x^{(0)}$ that minimizes the $\ell_2$ norm. Therefore, the following holds,

$$\|Ax^{(0)} - b\|_p^p \leq \|Ax^\star - b\|_p^p m^{(p-2)/2}.$$

The value of $i$ is the minimum at termination. Therefore, it is sufficient to prove the above bound for the termination condition. Our condition gives us the following at termination (the left inequality holds because otherwise, we would have terminated in the previous iteration).

$$i \geq \frac{\varepsilon}{16p(1+\varepsilon)}\|Ax - b\|_p^p \geq \frac{i}{2}.$$

This implies,

$$i \geq \frac{\varepsilon}{16p(1+\varepsilon)}\|Ax^\star - b\|_p^p \geq \frac{\varepsilon}{16p(1+\varepsilon)}\|Ax^{(0)} - b\|_p^p m^{-(p-2)/2}.$$

We next prove the second claim. Initially our solution satisfies $Cx^{(0)} = d$. Assuming the condition holds at iteration $t$, we show it for $t+1$. At every iteration we solve for $\widetilde{\Delta}$ under the constraint $C\widetilde{\Delta} = 0$. Our update rule, $x^{(t+1)} = x^{(t)} - \alpha\widetilde{\Delta}$ gives us,

$$Cx^{(t+1)} = Cx^{(t)} - \alpha C\widetilde{\Delta} = d - \alpha \cdot 0 = d.$$

$\square$

## B.4  Proof of Lemma 3.3

Our approximation depends on the quantity $\alpha_0$ which is defined in the algorithm. This depends on the value of $k$, the ratio of the $p$-norm term to the square term. Therefore, in order to bound the approximation, we first give a bound on $k$.

**Lemma B.6.** *Let $\widetilde{\Delta}$ the optimum of (4) and let $k = \frac{p^p \left\| A\widetilde{\Delta} \right\|_p^p}{2p^2 \widetilde{\Delta}^\top A^\top (R+sI) A\widetilde{\Delta}}$. If the optimum of (4) is at most $\lambda i/p^2$, then $k$ is at most $(32pm)^{(p-2)/2}$ for $\lambda = 16p$. Let $\alpha_0 = \min\left\{ \frac{1}{16\lambda}, \frac{1}{(16\lambda k)^{1/(p-1)}} \right\}$. Then $\alpha_0 \geq \Omega(p^{-1/2} m^{-\frac{(p-2)}{2(p-1)}})$ when $p \leq m$.*

*Proof.* Since, $sI \preceq R + sI$,

$$\left\| A\widetilde{\Delta} \right\|_2^2 = \widetilde{\Delta}^\top A^\top A\widetilde{\Delta} \leq \frac{1}{s} \widetilde{\Delta}^\top A^\top (R + sI) A\widetilde{\Delta}$$

and,

$$\left\| A\widetilde{\Delta} \right\|_p^p \leq \left\| A\widetilde{\Delta} \right\|_2^p \leq \frac{1}{s} \left( \widetilde{\Delta}^\top A^\top A\widetilde{\Delta} \right)^{(p-2)/2} \widetilde{\Delta}^\top A^\top (R + sI) A\widetilde{\Delta}.$$

We also have, $\widetilde{\Delta}^\top A^\top A\widetilde{\Delta} \leq \frac{2\lambda}{p^2} m^{(p-2)/p} i^{2/p}$. Combining these,

$$
\begin{aligned}
\frac{p^p \left\| A\widetilde{\Delta} \right\|_p^p}{2p^2 \widetilde{\Delta}^\top A^\top (R + sI) A\widetilde{\Delta}} &\leq \frac{p^p}{2p^2} \frac{2m^{(p-2)/p}}{i^{(p-2)/p}} \left( \frac{2\lambda}{p^2} m^{(p-2)/p} i^{2/p} \right)^{(p-2)/2} \\
&\leq (4\sqrt{2})^{p-2} p^{(p-2)/2} m^{(p-2)/p} m^{(p-2)^2/2p} \\
&= (32pm)^{(p-2)/2}
\end{aligned}
$$

We can now find a bound on $\alpha_0$.

$$
\begin{aligned}
\alpha_0 &\geq \min\left\{ \Omega\left( \frac{1}{p} \right), \Omega\left( \frac{1}{p^{1/(p-1)} (pm)^{(p-2)/2(p-1)}} \right) \right\} \\
&\geq \min\left\{ \Omega\left( \frac{1}{p} \right), \Omega\left( \frac{1}{p^{1/2} m^{(p-2)/2(p-1)}} \right) \right\} \\
&\geq \Omega\left( \frac{1}{p^{1/2} m^{(p-2)/2(p-1)}} \right), \text{ assuming } p \leq m.
\end{aligned}
$$

$\square$

**Lemma 3.3.** *(Approximation). Let $R, g, s, \alpha$ be as defined in lines (5), (6), (7) and (9) of Algorithm 1. Let $\alpha_0$ be as defined in line (4) of Algorithm 2 and $\widetilde{\Delta}$ be the solution of the following program,*

$$\arg\min_{\Delta} \Delta^\top A^\top (R + sI) A\Delta \quad s.t. \quad g^\top A\Delta = i/2, \, C\Delta = 0. \tag{4}$$

*If $\widetilde{\Delta}^\top A^\top (R+sI) A\widetilde{\Delta} \leq \lambda i/p^2$ and $\gamma(\alpha_0 \cdot \widetilde{\Delta}) \geq \frac{\alpha_0 i}{4}$, then $\alpha \cdot \widetilde{\Delta}$ is an $O\left( p^{1.5} m^{\frac{p-2}{2(p-1)}} \right)$- approximate solution to the residual problem.*

*Proof.* In the algorithm we choose $\alpha$ such that given $\widetilde{\Delta}$, $\alpha = argmin_\delta \| A(x - \delta\widetilde{\Delta}) - b \|_p^p$. From our assumption, we also know that $\gamma(\alpha_0 \widetilde{\Delta}) \geq \frac{\alpha_0}{4} i$. Now, since the residual function is a convex function

with value zero at the zero vector, we know that $\gamma(\widetilde{\Delta}/\lambda) \leq \frac{1}{\lambda}\gamma(\widetilde{\Delta})$ (for our value of $\lambda = 16p$).

$$
\begin{aligned}
\gamma(\alpha\widetilde{\Delta}) &\geq \lambda\gamma(\alpha\widetilde{\Delta}/\lambda) \\
&\geq \|\boldsymbol{A}\boldsymbol{x} - \boldsymbol{b}\|_p^p - \|\boldsymbol{A}(\boldsymbol{x} - \alpha\widetilde{\Delta}) - \boldsymbol{b}\|_p^p \\
&\geq \|\boldsymbol{A}\boldsymbol{x} - \boldsymbol{b}\|_p^p - \|\boldsymbol{A}(\boldsymbol{x} - \alpha_0\widetilde{\Delta}) - \boldsymbol{b}\|_p^p \\
&\geq \gamma(\alpha_0 \cdot \widetilde{\Delta}) \\
&\geq \frac{\alpha_0}{4}i \\
&\geq \frac{\alpha_0}{4\lambda}OPT.
\end{aligned}
$$

The last inequality follows form the fact that the optimum of the residual problem is at most $\lambda i$. This is because, Lemma 3.4 shows that, $(\|\boldsymbol{A}\boldsymbol{x}^{(t)} - \boldsymbol{b}\|_p^p - \|\boldsymbol{A}\boldsymbol{x}^\star - \boldsymbol{b}\|_p^p)/16p < i$. Now from Lemma B.2 we can conclude that the residual problem has optimum at most $\lambda i$. Since we have from our assumption that the objective of (4) is at most $\lambda i/p^2$, from Lemma B.6, we can bound the factor $O(\lambda/\alpha_0) \leq O(p^{1 + \frac{p-2}{2(p-1)}} m^{\frac{p-2}{2(p-1)}}) \leq O(p^{1.5} m^{\frac{p-2}{2(p-1)}})$.

$\square$

## B.5 Proof of Lemma 3.5

**Lemma 3.5.** *(Termination). Let $i$ be such that $(\|\boldsymbol{A}\boldsymbol{x}^{(t)} - \boldsymbol{b}\|_p^p - \|\boldsymbol{A}\boldsymbol{x}^\star - b\|_p^p)/16p \in (i/2, i]$. Then,*

$$
i \leq \frac{\varepsilon}{16p(1+\varepsilon)}\|\boldsymbol{A}\boldsymbol{x}^{(t)} - \boldsymbol{b}\|_p^p \Rightarrow \|\boldsymbol{A}\boldsymbol{x}^{(t)} - \boldsymbol{b}\|_p^p \leq (1+\varepsilon)\mathrm{OPT}.
$$

*and,*

$$
\|\boldsymbol{A}\boldsymbol{x}^{(t)} - \boldsymbol{b}\|_p^p \leq (1+\varepsilon)\mathrm{OPT} \Rightarrow i \leq 2\frac{\varepsilon}{16p(1+\varepsilon)}\|\boldsymbol{A}\boldsymbol{x}^{(t)} - \boldsymbol{b}\|_p^p.
$$

*Proof.* We first show the forward implication. From the assumptions, we have,

$$
\frac{\left\|\boldsymbol{A}\boldsymbol{x}^{(t)} - \boldsymbol{b}\right\|_p^p - OPT}{16p} \leq i \leq \frac{\varepsilon}{16p(1+\varepsilon)}\left\|\boldsymbol{A}\boldsymbol{x}^{(t)} - \boldsymbol{b}\right\|_p^p
$$

$$
\Rightarrow \left\|\boldsymbol{A}\boldsymbol{x}^{(t)} - \boldsymbol{b}\right\|_p^p \frac{1}{1+\varepsilon} \leq OPT
$$

$$
\Rightarrow \left\|\boldsymbol{A}\boldsymbol{x}^{(t)} - \boldsymbol{b}\right\|_p^p \leq (1+\varepsilon)OPT.
$$

For the other direction we have,

$$
\frac{\left\|\boldsymbol{A}\boldsymbol{x}^{(t)} - \boldsymbol{b}\right\|_p^p}{1+\varepsilon} \leq OPT.
$$

Thus,

$$
i \leq 2\frac{\left\|\boldsymbol{A}\boldsymbol{x}^{(t)} - \boldsymbol{b}\right\|_p^p - OPT}{16p} \leq 2\frac{\left\|\boldsymbol{A}\boldsymbol{x}^{(t)} - \boldsymbol{b}\right\|_p^p}{16p}\left(1 - \frac{1}{1+\varepsilon}\right) \leq \frac{2\varepsilon}{1+\varepsilon}\frac{\left\|\boldsymbol{A}\boldsymbol{x}^{(t)} - \boldsymbol{b}\right\|_p^p}{16p}.
$$

$\square$

# Appendix C   Converting $\ell_p$-Laplacian Minimization to Regression Form

Define the following terms:

- $n$ denote the number of vertices.
- $l$ denote the number of labels.
- $\boldsymbol{B}$ denote the edge-vertex adjacency matrix.

- $g$ denote the vector of labels for the $l$ labelled vertices.
- $W$ denote the diagonal matrix with weights of the edges.

Set $A = W^{1/p}B$ and $b = -B[:, n : n + l]g$. Now $\|Ax - b\|_p^p$ is equal to the $\ell_p$ laplacian and we can use our IRLS algorithm to find the $x$ that minimizes this.

## Appendix D  Solving $\ell_2$ Problems under Subspace Constraints

### D.1  Finding the Initial Solution

We want to solve:

$$\min_{x} \quad \|Ax - b\|_2^2$$
$$Cx = d.$$

Using Lagrangian duality and noting that strong duality holds, we can write the above as,

$$L(x, v) = \min_{x} \max_{v} \quad (Ax - b)^\top (Ax - b) + v^\top (d - Cx)$$
$$= \max_{v} \min_{x} \quad (Ax - b)^\top (Ax - b) + v^\top (d - Cx).$$

We first find $x^\star$ that minimizes the above objective by setting the gradient with respect to $x$ to $0$. We thus have,

$$x^\star = (A^\top A)^{-1} \left( \frac{2A^\top b + C^\top v}{2} \right).$$

Using this value of $x$ we arrive at the following dual program.

$$L(v) = \max_{v} \quad -\frac{1}{4}v^\top C(A^\top A)^{-1}C^\top v - b^\top A(A^\top A)^{-1}A^\top b - v^\top C(A^\top A)^{-1}A^\top b + b^\top b + v^\top d,$$

which is optimized at,

$$v^\star = 2\left( C(A^\top A)^{-1}C^\top \right)^{-1} \left( d - C(A^\top A)^{-1}A^\top b \right).$$

Strong duality also implies that $L(x, v^\star)$ is optimized at $x^\star$, which gives us,

$$x^\star = (A^\top A)^{-1} \left( A^\top b + C^\top \left( C(A^\top A)^{-1}C^\top \right)^{-1} \left( d - C(A^\top A)^{-1}A^\top b \right) \right).$$

### D.2  Solving (4)

At every iteration of the algorithm, we want to solve the following problem,

$$\min_{\Delta} \quad \Delta^\top A^\top (R + sI)A\Delta$$
$$g^\top A\Delta = i/2$$
$$C\Delta = 0.$$

The constraints can be combined and rewritten as, $C'\Delta = d'$ where,

$$C' = \begin{bmatrix} C \\ g^\top A \end{bmatrix}, \quad d' = \begin{bmatrix} 0 \\ i/2 \end{bmatrix}.$$

Let $R' = R + sI$. We now want to solve,

$$\min_{\Delta} \quad \|R'^{1/2}A\Delta\|_2^2$$
$$C'\Delta = d'.$$

Using a procedure similar as in the previous section, we get,

$$\Delta^\star = (A^\top R'A)^{-1}C'^\top \left( C'(A^\top R'A)^{-1}C'^\top \right)^{-1} d'$$