[Reviews · NeurIPS 2019]

Reviewer 1



The paper proposes a simple algorithm for L_p regression problems and justifies its efficiency both theoretically and empirically. Major concerns: 1) Related work: The comparison with related works is not sufficient. [AKPS19] has a (log(1/\epsilon))^2 dependence in the number of iterations, but it is not mentioned in Section 1.1. In Section, it only compares with the polynomial dependence. 2) As mentioned in the related work, the dependence on m of the proposed method is marginally worse than Bubeck et al. [BCLL19] and Adil et al. [AKPS19] but the overall comparisons are not clear. What are the advantages of the proposed one? Furthermore, the lack of sufficient baselines (at least Bubeck et al. [BCLL19] or Adil et al. [AKPS19]) in the numerical experiments weakens the superior of the proposed algorithm. 3) This paper needs significant rewriting. One example is the logic in the first paragraph of page 2 (line [34] to [40]). It first says that the convergence in solution is more important than function values, and then argued that this paper will measure the convergence in the function value. Figure 4 needs more spaces between subfigures to avoid confusion. ~~~~~~~~~~~~~~~~~~~~~~~~~~~~~~~~~~~~~~~~~~~~~~~~~~~~~~~~~~~~~~ The authors answered several questions regarding the comparison with existing methods. They are mainly on the theoretical side. The numerical part is difficult because there is no available codes. Theoretical results are mostly based on the worst case, so the referee is not fully convinced by the response on the numerical part. Still this is a good submission, and the overall score will not be changed.

Reviewer 2



The paper proposes a novel IRLS method to solve l_p linear regression for p>=2. The method is the first IRLS algorithm with provable linear convergence to the optimum. The paper is clearly written and well structured and easy to read. The obtained results (Thm. 3.1) are currently state-of-the-art for the given problem class. Furthermore, the experiments are reasonable. Although I did not go through the proofs, the obtained results are believable. Overall, I do not have any objection against the paper, and I think it is in good shape to be accepted. However, I must say I am not an expert on the IRLS algorithms. Minor comments: 1) l. 99: p-IRLS is said to be far simpler to implement to methods mentioned in the previous paragraph. This is not completely true; as the method from Maddison et al is in fact way simpler to implement to p-IRLS. Furthermore, the method from Maddison et al also allows for the linesearch. I suggest the authors make the mentioned paragraph more clear. 2) The author names in the references are inconsistent, some have fist name shortened while others do not

Reviewer 3



This paper is clearly written, well-motivated, and, to the best of this reviewer's knowledge, the proposed algorithm and corresponding analysis are novel. Of course, one may claim that the level of originality is limited, given that the proposed algorithm is nothing more than the combination of a couple of improvements to IRLS that had been independently proposed before, but the way in which they are combined and the corresponding analysis is novel. Moreover, the resulting algorithm is extremely fast in practice, as shown in the experimental section of the paper. -------------------- After reading the other reviews and the authors' response, I see no reason to change my score (8).

[Author Response · NeurIPS 2019]

We thank all the reviewers for their helpful comments and suggestions. Below we address the concerns raised.

**Importance of Convergence vs Function value (R1).** For an algorithm with a $\log \frac{1}{\varepsilon}$ dependence of the running time
for computing a $(1 + \varepsilon)$-approximate solution, like $p$-IRLS, the guarantee can be translated into a guarantee for
convergence in the solution without any significant loss in the runtime complexity of the method. We demonstrate this
theoretically and experimentally below. We thank the reviewer for pointing out that this is inadequately explained in the
paper, and we will clarify this in the final version of the paper.

If $\boldsymbol{x}$ is a $(1 + \delta)$-approximate solution, using Lemma A.1 from the supplementary material we can show that
we can achieve the guarantee $\|\boldsymbol{x} - \boldsymbol{x}^\star\|_\infty \leq \varepsilon \|\boldsymbol{A}\boldsymbol{x}^\star - \boldsymbol{b}\|_p$ by picking $\delta = \left(\frac{\varepsilon \sigma_{\min}(\boldsymbol{A})}{4m}\right)^p$, where $\sigma_{\min}(\boldsymbol{A})$ is
the smallest singular value of $\boldsymbol{A}$. This gives $\log \frac{m}{\delta} = O(p \log \frac{m}{\sigma_{\min}(\boldsymbol{A})\varepsilon})$, and hence a total iteration count of
$O(p^{4.5} m^{\frac{p-2}{2(p-1)}} \log \frac{m}{\sigma_{\min}(\boldsymbol{A})\varepsilon})$. Asymptotically, the running time bound is only off by a factor of $p$ if we wish to
measure the convergence in $\ell_\infty$-norm, as long as $\log \frac{1}{\sigma_{\min}(\boldsymbol{A})} = O(\log \frac{m}{\varepsilon})$.

Error: Matrix

Error: Graphs

We also demonstrate this relation experimentally. The plots demonstrate the
average resulting $\ell_\infty$ norm deviation for the solution computed, as we change the
$\varepsilon$ parameter used in the algorithm. We use the instances described in the paper;
matrices of size $1000 \times 800$ and graphs with $1000$ nodes. For each instance, we: 1)
find a very high accuracy solution, by choosing a very small $\varepsilon \sim 10^{-25}$, 2) scale
the problem so that the optimum value is 1, and run the algorithm again to find the
optimum solution $\boldsymbol{x}^\star$. 3) Now we have a problem such that $\|\boldsymbol{A}\boldsymbol{x}^\star - \boldsymbol{b}\|_p = 1$,
we run the algorithm again with various values of $\varepsilon$, to obtain solutions $\boldsymbol{x}(\varepsilon)$ and
plot $\|\boldsymbol{x}(\varepsilon) - \boldsymbol{x}^\star\|_\infty$ (averaged over 20 samples). These results are very much in
agreement with the theoretical $\varepsilon^{\frac{1}{p}}$ dependence proved above. (Note that the error
bars indicate $\log(\text{mean} \pm \text{std})$ so they are missing on one side when mean $<$ std.)

**Runtime comparison with [AKPS19] and [BCLL18] (R1).** As noted by R1,
the running time of [AKPS19] (and [BCLL18]) is not stated precisely in the
comparison. The running time bounds are not stated precisely in either paper;
they hide the $p$ dependencies and $\text{poly}(\log \frac{m}{\varepsilon})$ dependencies. We have focused
on the polynomial terms in the comparison because they are the dominant terms.
For [AKPS19] the running time is at least $p^{2p+2} m^{\frac{p-2}{3p-2}} \log^2 \frac{m}{\varepsilon}$, for [BCLL18] it
seems to be at least $p^{2.5} m^{\frac{p-2}{2p}} \log^2 \frac{m}{\varepsilon}$. The $\log^2 \frac{m}{\varepsilon}$ dependence is worse for both
[AKPS19] and [BCLL18], compared to our algorithm, and the $p^{2p+2}$ factor is
much worse in [AKPS19]. We will clarify this in the paper.

**Experimental comparison to [AKPS19] and [BCLL18] (R1).** We agree that a direct comparison to [AKPS19] and
[BCLL18] is desirable. Unfortunately, both algorithms are quite complicated to implement, and no implementations are
publicly available. The [BCLL18] paper lacks an explicit algorithm description and leaves out several details (e.g. it
asks to run accelerated gradient descent (AGD) "until convergence", the specific accuracy target for AGD will have a
large impact on the running time). The [AKPS19] algorithm description also leaves out specifying several parameters
in the algorithm, hiding $p$ dependencies and $\log \frac{m}{\varepsilon}$ factors. As pointed out above, these large hidden factors make the
algorithm, as stated, difficult to implement efficiently. In contrast, our algorithm is far simpler to implement.

**Simplicity of $p$-IRLS compared to [MPT+18] (R3).** We thank R3 for this. We will clarify this in the final version.

**Combining $p$-norm with a regularizer e.g. $\ell_1$ (Lasso) (R3).** This is definitely a great idea for future work. Our
current techniques would not suffice for this, but we thank the reviewer for pointing out this potential direction.

**Spacing between subfigures in figure 4 (R1)** We will address this in the final version.

**Proof of claimed bound.** We prove the bound on $\|\boldsymbol{x} - \boldsymbol{x}^\star\|_\infty$ claimed above. Given that $\boldsymbol{x}$ is a $(1 + \delta)$-approximate
solution, using Lemma A.1, we can write the following lower bound on the objective value:

$$(1 + \delta) \|\boldsymbol{A}\boldsymbol{x}^\star - \boldsymbol{b}\|_p^p \geq \|\boldsymbol{A}\boldsymbol{x}^\star - \boldsymbol{b}\|_p^p + p \left(\boldsymbol{A}\boldsymbol{x}^\star - \boldsymbol{b}\right)^\top \boldsymbol{R}\boldsymbol{A}(\boldsymbol{x} - \boldsymbol{x}^\star) + \frac{p}{8}\boldsymbol{A}(\boldsymbol{x} - \boldsymbol{x}^\star)^\top \boldsymbol{A}^\top \boldsymbol{R}\boldsymbol{A}(\boldsymbol{x} - \boldsymbol{x})^\star + 2^{-(p+1)} \|\boldsymbol{A}\boldsymbol{x} - \boldsymbol{A}\boldsymbol{x}^\star\|_p^p,$$

where $\boldsymbol{R} = \text{diag}(|\boldsymbol{A}\boldsymbol{x}^\star - \boldsymbol{b}|^{p-2})$. Since the gradient at $\boldsymbol{x}^\star$ is 0, simplifying, we get, $2^{p+1}\delta \|\boldsymbol{A}\boldsymbol{x}^\star - \boldsymbol{b}\|_p^p \geq$
$\|\boldsymbol{A}\boldsymbol{x} - \boldsymbol{A}\boldsymbol{x}^\star\|_p^p$. Now, translating between various norms, we obtain,

$$\|\boldsymbol{x} - \boldsymbol{x}^\star\|_\infty \leq \frac{1}{\sigma_{\min}(\boldsymbol{A})} \|\boldsymbol{A}\boldsymbol{x} - \boldsymbol{A}\boldsymbol{x}^\star\|_2 \leq \frac{m^{\frac{1}{2} - \frac{1}{p}}}{\sigma_{\min}(\boldsymbol{A})} \|\boldsymbol{A}\boldsymbol{x} - \boldsymbol{A}\boldsymbol{x}^\star\|_p \leq \frac{2m^{\frac{1}{2}}}{\sigma_{\min}(\boldsymbol{A})} \left(\frac{2\delta}{m}\right)^{\frac{1}{p}} \|\boldsymbol{A}\boldsymbol{x}^\star - \boldsymbol{b}\|_p.$$

[Meta-Review · NeurIPS 2019]

This paper proposes a modified IRLS algorithm and presents empirical experiments and theoretical analysis. The reviewers viewed the contribution as a combination of existing methods, and the combination is novel. Most reviewers thought the paper was well-written. The ability of the authors to compare to existing methods is limited because of the complexity of other methods and lack of public implementations. The reviewers' scores place this paper above the bar for acceptance.